# Completion of isoniazid–rifapentine (3HP) for tuberculosis prevention among people living with HIV: Interim analysis of a hybrid type 3 effectiveness–implementation randomized trial

Fred C. Semitala[1,2,3‡], Jillian L. Kadota[4‡], Allan Musinguzi[2], Juliet Nabunje[2], Fred Welishe[2], Anne Nakitende[2], Lydia Akello[2], Opira Bishop[2], Devika Patel[5], Amanda Sammann[5], Payam Nahid[4], Robert Belknap[6], Moses R. Kamya[1,2], Margaret A. Handley[7,8], Patrick P. J. Phillips[4], Anne Katahoire[9], Christopher A. Berger[4], Noah Kiwanuka[10], Achilles Katamba[10,11], David W. Dowdy[11,12‡], Adithya Cattamanchi[4,11‡*]

1 Makerere University, Department of Medicine, College of Health Sciences, Kampala, Uganda, 2 Infectious Diseases Research Collaboration, Kampala, Uganda, 3 Makerere University Joint AIDS Program, Kampala, Uganda, 4 UCSF Center for Tuberculosis and Division of Pulmonary and Critical Care Medicine, San Francisco General Hospital, University of California, San Francisco, San Francisco, California, United States of America, 5 The Better Lab, Department of Surgery, University of California, San Francisco, San Francisco, California, United States of America, 6 Denver Health and Hospital Authority and Division of Infectious Diseases, Department of Medicine, University of Colorado, Denver, Colorado, United States of America, 7 Center for Vulnerable Populations at Zuckerberg San Francisco General Hospital and Trauma Center, University of California, San Francisco, San Francisco, California, United States of America, 8 Department of Epidemiology and Biostatistics, University of California, San Francisco, San Francisco, California, United States of America, 9 Child Health and Development Center, School of Medicine, Makerere University College of Health Sciences, Kampala, Uganda, 10 Clinical Epidemiology & Biostatistics Unit, Department of Medicine, Makerere University College of Health Sciences, Kampala, Uganda, 11 Uganda Tuberculosis Implementation Research Consortium, Kampala, Uganda, 12 Department of Epidemiology, Johns Hopkins Bloomberg School of Public Health, Baltimore, Maryland, United States of America

‡ These authors contributed equally as first authors to this work. AC and DD also contributed equally as senior authors to this work.
* adithya.cattamanchi@ucsf.edu

## Abstract

### Background

Scaling up shorter regimens for tuberculosis (TB) prevention such as once weekly isoniazid–rifapentine (3HP) taken for 3 months is a key priority for achieving targets set forth in the World Health Organization's (WHO) END TB Strategy. However, there are few data on 3HP patient acceptance and completion in the context of routine HIV care in sub-Saharan Africa.

### Methods and findings

The 3HP Options Trial is a pragmatic, parallel type 3 effectiveness–implementation randomized trial comparing 3 optimized strategies for delivering 3HP—facilitated directly observed

**Data Availability Statement:** All relevant data are within the manuscript and its Supporting information files.

**Funding:** This study is supported by a grant from the US National Heart, Lung and Blood Institute: NIH/NHLBI R01HL144406 (AC); https://www.nhlbi.nih.gov/. The funders had no role in study design, data collection and analysis, decision to publish, or preparation of the manuscript.

**Competing interests:** I have read the journal's policy and the authors of this manuscript have the following competing interests: AS receives research funding from Global Health Labs, California Health Care Foundation and is owner of The Empathy Studio, LLC. DP is a human-centered design consultant for The Empathy Studio, LLC. The other authors have declared that no competing interests exist.

**Abbreviations:** AE, adverse event; ALT, alanine aminotransferase; ART, antiretroviral therapy; AST, aspartate transaminase; CI, confidence interval; DAT, digital adherence technology; DOT, directly observed therapy; ISS, Immune Suppression Syndrome; IVR, interactive voice response; MPI, Multidimensional Poverty Index; PEPFAR, President's Emergency Plan for AIDS Relief; PLHIV, people living with HIV; SAT, self-administered therapy; SD, standard deviation; SMS, short messaging service; TB, tuberculosis; TPT, tuberculosis preventive therapy; ULN, upper limit of normal; WHO, World Health Organization.

therapy (DOT), facilitated self-administered therapy (SAT), or informed choice between DOT and SAT using a shared decision-making aid—to people receiving care at a large urban HIV clinic in Kampala, Uganda. Participants and healthcare providers were not blinded to arm assignment due to the nature of the 3HP delivery strategies. We conducted an interim analysis of participants who were enrolled and exited the 3HP treatment period between July 13, 2020 and April 30, 2021. The primary outcome, which was aggregated across trial arms for this interim analysis, was the proportion who accepted and completed 3HP ($\geq$11 of 12 doses within 16 weeks of randomization). We used Bayesian inference analysis to estimate the posterior probability that this proportion would exceed 80% under at least 1 of the 3HP delivery strategies, a coprimary hypothesis of the trial. Through April 2021, 684 participants have been enrolled, and 479 (70%) have exited the treatment period. Of these 479 participants, 309 (65%) were women, mean age was 41.9 years (standard deviation (SD): 9.2), and mean time on antiretroviral therapy (ART) was 7.8 years (SD: 4.3). In total, 445 of them (92.9%, 95% confidence interval (CI): [90.2 to 94.9]) accepted and completed 3HP treatment. There were no differences in treatment acceptance and completion by sex, age, or time on ART. Treatment was discontinued due to a documented adverse event (AE) in 8 (1.7%) patients. The probability that treatment acceptance and completion exceeds 80% under at least 1 of the three 3HP delivery strategies was greater than 99%. The main limitations are that the trial was conducted at a single site, and the interim analysis focused on aggregate outcome data to maintain blinding of investigators to arm-specific outcomes.

## Conclusions

3HP was widely accepted by people living with HIV (PLHIV) in Uganda, and very high levels of treatment completion were achieved in a programmatic setting. These findings show that 3HP can enable effective scale-up of tuberculosis preventive therapy (TPT) in high-burden countries, particularly when delivery strategies are tailored to target known barriers to treatment completion.

## Trial registration

ClinicalTrials.gov NCT03934931.

## Author summary

### Why was this study done?

- Short course (12 weeks) isoniazid–rifapentine (3HP) is safe and effective for preventing tuberculosis (TB) among people living with HIV (PLHIV) and was recently recommended by the World Health Organization (WHO).

- However, data on successful implementation (i.e., high acceptance and completion) in high-burden settings are very limited.

- Evidence of successful implementation is critical for informing country-level decisions to scale up 3HP.

**What did the researchers do and find?**

- We conducted an interim analysis of 479 participants in an ongoing randomized trial of 3 strategies for 3HP delivery taking place at a large, urban HIV/AIDS clinic in Kampala, Uganda.

- All aspects of 3HP treatment—including counseling, drug administration, side effect monitoring, and adherence evaluation—were done by routine healthcare providers.

- Acceptance and completion of 3HP was high (93%, with >99% probability of exceeding 80% in at least 1 trial arm). Only 8 (1.7%) patients discontinued 3HP treatment due to an adverse event (AE).

**What do these findings mean?**

- High levels of 3HP acceptance and completion can be achieved in the context of routine HIV/AIDS care in sub-Saharan Africa.

- HIV/AIDS programs in sub-Saharan Africa should consider scaling up 3HP as an alternative to isoniazid preventive therapy.

- Further research is needed to confirm that the facilitation strategies evaluated in the 3HP Options Trial are feasible in other settings and needed to enhance treatment acceptance and completion.

## Introduction

Tuberculosis (TB) is curable and preventable, yet remains a leading cause of death among people living with HIV (PLHIV) [1]. Tuberculosis preventive therapy (TPT) can reduce TB incidence and mortality by 30% to 50% [2,3] and is recommended for all PLHIV in high TB burden settings. Access to TPT has improved recently, with 3.5 million PLHIV receiving TPT in 2019 [1]. However, in many settings, less than half of those initiating TPT with the standard 6-month course of daily isoniazid complete treatment [4–6].

Safer, shorter TPT regimens are now available, including a 12-dose weekly regimen of isoniazid–rifapentine (3HP) [7–9]. In 2018, the World Health Organization (WHO) recommended 3HP as an option for TPT, citing its shortened duration as an enabler of treatment adherence and completion [1]. The Phase IV multicenter iAdhere trial conducted primarily in the United States found that 3HP treatment completion was 87.2% with directly observed therapy (DOT), 74% with self-administered therapy (SAT), and 76.4% with SAT plus short messaging service (SMS) reminders [10]. However, data on acceptance and completion of 3HP from high-burden settings—particularly in sub-Saharan Africa—remain sparse, posing a major barrier to widespread uptake and implementation [11,12]. Notably, in the iAdhere trial, noninferiority of SAT to DOT was only demonstrated among participating sites in the US, with SAT performing worst (37% treatment completion) in South Africa, the only participating African site. These findings further underscore the need for evidence of 3HP acceptability

and completion in high-burden settings like Uganda. Under the recommendation of our external Trial Steering Committee and taking into account the urgent need for data from high-burden settings to support country-level decisions regarding 3HP scale-up, we undertook an interim analysis of the 3HP Options Trial, a randomized trial of 3 facilitated strategies for delivering 3HP to PLHIV receiving routine HIV/AIDS care in Uganda.

## Methods

### Study design and participants

The 3HP Options Trial is an ongoing 3-arm, parallel, individual participant randomized trial with a hybrid type 3 effectiveness–implementation design. Hybrid effectiveness–implementation trials assess both effectiveness and implementation outcomes, with type I designs having a greater focus on intervention effectiveness and type 3 designs having a greater focus on implementation strategies and outcomes [13]. Our overall goal as a type 3 hybrid design is to identify the optimal implementation strategy (facilitated DOT, facilitated SAT, or patient choice) for enabling completion of 3HP (i.e., the evidence-based intervention), while also adding to the large body of evidence demonstrating its safety and effectiveness. The trial is being conducted among adults accessing HIV/AIDS care at the Mulago Immune Suppression Syndrome (ISS; i.e., HIV/AIDS) clinic in Kampala, Uganda. This clinic is the largest outpatient HIV clinic in Uganda (over 16,000 PLHIV enrolled and 300 new PLHIV registered monthly), and scale-up of TPT is a key priority for clinic leadership as well as the Uganda National Tuberculosis and Leprosy Programme and AIDS Control Programme at the Uganda Ministry of Health.

Details of the trial design—including full eligibility criteria—have been published previously [14]. Briefly, the trial included adults (age ≥18 years) living with HIV engaged in care at the Mulago ISS clinic. PLHIV were excluded if they were ineligible for 3HP treatment (e.g., a baseline alanine aminotransferase [ALT] level or aspartate transaminase [AST] level >3 times the upper limit of normal) or had a logistical issue that impeded participation in the trial (e.g., unable to provide consent) (Fig 1).

Trial enrollment began in July 2020 and is anticipated to be completed in October 2022. The trial was approved by institutional review boards at Makerere University School of Public Health Higher Degrees, Research and Ethics Committee, the University of California, San Francisco Human Research Protection Program, and by the Uganda National Council for Science and Technology. The trial is registered on ClinicalTrials.gov (NCT03934931). Plans to report an interim analysis were reviewed and approved by the independent Trial Steering Committee.

### Randomization and interventions

Briefly, eligible participants who provide written informed consent are randomized to 1 of 3 optimized delivery strategies for receiving 3HP. The delivery strategies were designed to address barriers to 3HP acceptance and completion identified in formative research [15–17] and with involvement of local stakeholders including PLHIV, clinicians, and clinic/program leadership to maximize acceptability and feasibility.

The trial protocol paper provides full details of the components of the 3 delivery strategies: facilitated DOT, facilitated SAT, and patient choice between facilitated DOT and facilitated SAT (with the assistance of a decision aid) [14]. Briefly, all strategies included (a) standardized patient counseling about the risks and benefits of 3HP; (b) streamlined clinic visits (11 visits for DOT and 2 for SAT) at the pharmacy window for symptom-based TB screening and side effects screening using a standardized checklist (S1 Text); (c) reimbursement of transport costs for clinic visits; and (d) automated interactive voice response (IVR) reminders for clinic visits. The transport reimbursement was provided by pharmacy technicians with the amount

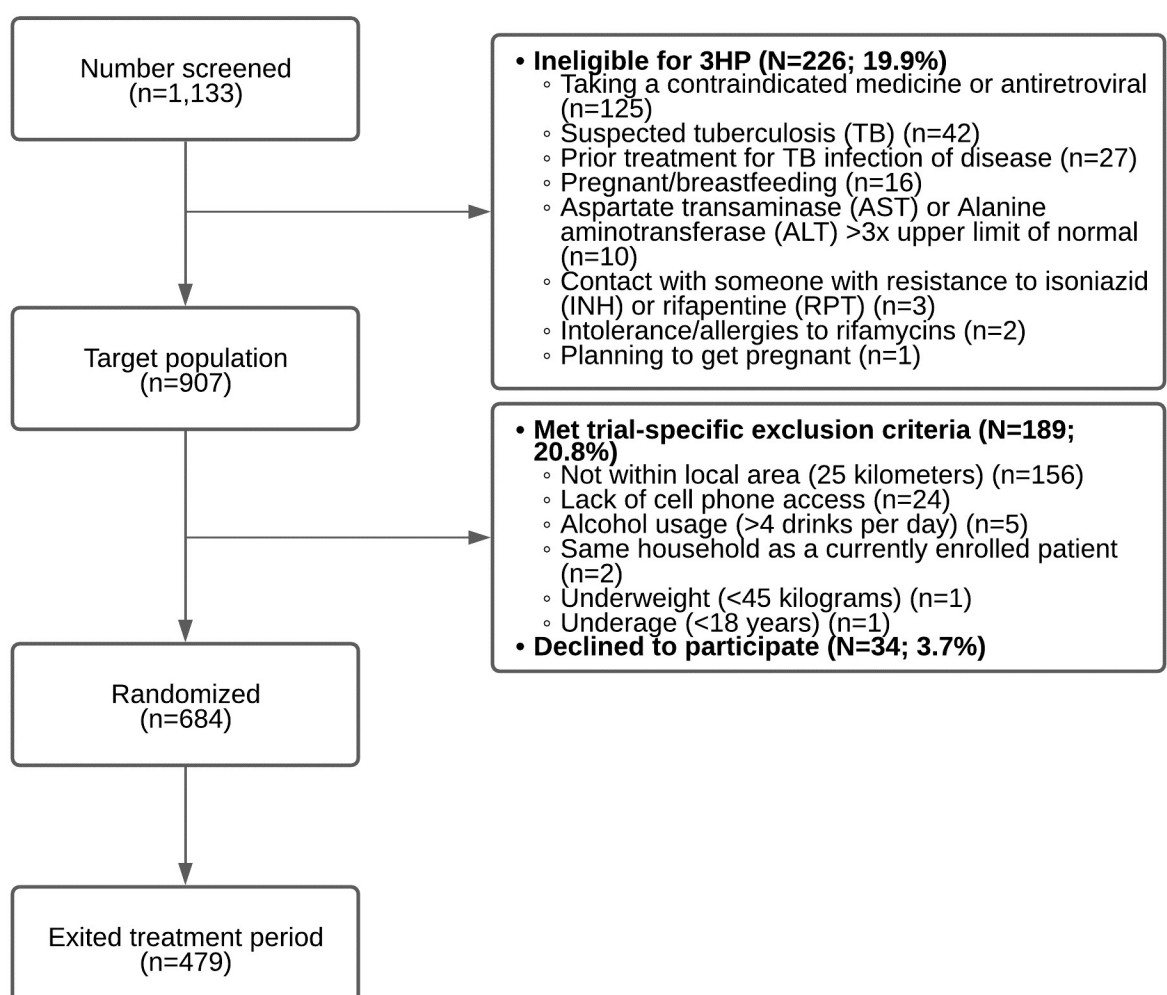

**Fig 1. CONSORT flow diagram.** *3HP Options Trial screening*, randomization, and allocation, July 13, 2020 to April 30, 2021 (*n* = 1,133). ALT, alanine aminotransferase; AST, aspartate transaminase; CONSORT, Consolidated Standards of Reporting Trials; INH, isoniazid; RPT, rifapentine; TB, tuberculosis.

standardized based on an assessment of average local transportation costs and stakeholder assessment of programmatic feasibility at scale. For participants randomized to the facilitated DOT arm, all 3HP doses were provided at weekly clinic visits. For participants randomized to the facilitated SAT arm, the first, sixth, and final 3HP doses were directly observed at clinic visits. The remaining doses were self-administered, and participants were asked to report dosing completion using the 99DOTS platform (Everwell Health Solutions, India). 99DOTS is a low-cost digital adherence technology wherein patients call random toll-free numbers prepackaged with each medication dose, and the calls are logged on the 99DOTS server to generate a real-time dosing history that health workers can monitor [18]. The 99DOTS platform was also used to send automated weekly IVR check-in calls asking "Are you well?" to assess for potential adverse events (AEs). For participants randomized to the choice arm, a decision aid was used to offer an informed choice between the facilitated DOT and facilitated SAT strategies. The decision aid was codesigned with PLHIV to help patients prioritize their values and choose the option that best aligned with their preferences. Participants in the choice arm could switch delivery strategies at any time.

## Assessments and data collection

Research staff screened PLHIV for eligibility, obtained informed consent from eligible participants, administered a baseline demographic and clinical survey, and performed randomization. Participants were then transferred to routine healthcare providers who performed all activities related to 3HP counseling, AE monitoring, dosing and adherence monitoring, and follow-up [14]. Healthcare providers did not receive any incentives or payments for delivering care related to 3HP. For AE monitoring, pharmacy technicians were asked to screen participants for potential side effects using a standardized checklist at clinic visits prior to 3HP dosing or by calling SAT participants who responded "No" to the weekly IVR check-in and to refer participants with potential side effects to a routine clinician for further evaluation. Further assessment including any laboratory testing was at the discretion of the clinician. Completion was assessed using the 99DOTS platform (i.e., weekly confirmation of self-administered doses by patient phone calls, with clinic-administered doses logged by pharmacy technicians on patients' behalf). Self-administered doses were further verified by pharmacy technician pill count. Pharmacy technicians were asked to review participant dosing histories and call participants who missed clinic visits or did not call the 99DOTS platform to report expected self-administered doses.

To extract AE-related data, research staff retrospectively reviewed (on a weekly basis) the AEs checklist completed by pharmacy technicians and clinic charts of all participants referred to a clinician for further evaluation of a potential AE. Research staff noted whether the clinician documented the potential AE to be 3HP related and whether 3HP treatment was continued, held, or discontinued permanently. To assess 3HP initiation and completion among participants in all 3 trial arms, research staff extracted weekly 3HP dosing data from the 99DOTS server.

## Trial outcomes

Trial outcomes were developed using the RE-AIM evaluation framework [19]. The primary outcome is the proportion of randomized participants who accept 3HP treatment and complete at least 11 of the 12 weekly doses within 16 weeks of enrollment [7,9,10], which reflects intervention reach (acceptance) and fidelity (treatment completion). In addition, this interim analysis assessed the proportion of participants initiating 3HP treatment who had a serious AE, defined as any AE resulting in treatment discontinuation, which reflects the effectiveness domain of the RE-AIM framework. Additional trial effectiveness and implementation outcomes are described in the published trial protocol [14] and will be reported upon completion of the trial.

## Statistical analysis

Our interim analysis included simple calculation (using exact binomial 95% confidence intervals (CIs)) of outcomes aggregated across trial arms overall and stratified by sex, age, and time on ART. We used Bayesian inference to estimate the posterior probability that the proportion of PLHIV who accept and complete 3HP would exceed 0.8 under at least 1 of the 3 delivery strategies being compared (a coprimary trial hypothesis, as prespecified in the trial protocol). We used a beta distribution for the primary outcome and used a noninformative flat conjugate beta (1,1) prior. We conservatively assumed that outcomes were distributed equally across the 3 study arms; if outcomes were distributed unequally, the probability of the outcome proportion exceeding 0.8 would be higher in at least 1 arm. We then calculated the posterior probability that the proportion accepting and completing 3HP in a single arm exceeded 0.8 using standard methods taking advantage of the conjugate beta prior [20].

## Results

From July 2020 through April 2021, 1,133 PLHIV accessing care at Mulago ISS clinic were screened for participation, and 226 (20%) had a contraindication to 3HP. Of the remaining 907 PLHIV who were in the target population for 3HP-based TPT, 223 (25%) were excluded for trial-specific reasons. Thus, 684 participants were randomized, and 479 participants who had exited the 16-week treatment period were included in the interim analysis (Fig 1). Of these 479 participants, 309 (65%) were women, mean age was 41.9 years (standard deviation (SD): 9.2), and mean time on antiretroviral therapy (ART) was 7.8 years (SD: 4.3; Table 1).

**Table 1. Baseline demographic and clinical characteristics.** 3HP Options Trial participants, July 13, 2020 to April 30, 2021 (*n* = 479).

| Total randomized | N = 479 |
|---|---|
| **Clinical** | **Mean (SD), *n* (%)** |
| Age | 41.9 (9.2) |
| Female sex | 309 (64.5%) |
| Prior TB | 90 (18.8%) |
| On ART | 479 (100.0%) |
| Years on ART | 7.8 (4.3) |
| Most recent CD4 count | 1,223 (2,344) |
| BMI (4 participants missing BMI) | 26.1 (5.4) |
| Typical travel time to clinic (hours) | 1.1 (0.7) |
| **Sociodemographic** | |
| **Employment status** | |
| Student | 5 (1.0) |
| Not employed | 57 (11.9) |
| Self-employed | 245 (51.2) |
| Hired worker | 47 (9.8) |
| Temporary/informal work | 125 (26.1) |
| **Education level** | |
| None | 41 (8.6) |
| Primary | 222 (46.4) |
| Secondary | 175 (36.5) |
| Vocational training/tertiary | 24 (5.0) |
| Postsecondary | 17 (3.6) |
| **MPI category**[1] | |
| Not vulnerable to poverty | 180 (37.6) |
| Not poor but vulnerable to poverty | 182 (38.0) |
| Multidimensionally poor | 90 (18.8) |
| Severely multidimensionally poor | 27 (5.6) |

[1]. The global MPI examines deprivations across 10 indicators in dimensions of health, education, and standards of living, with those deprived in one-third or more of the 10 indicators counted as being multidimensionally poor [21]. Health and education indicators are weighted at 1/6 each and standard of living indicators are weighted 1/18 each. MPI values can range from 0 to 1, with greater values indicating higher poverty. Variables included in our calculation of MPI category included (1) child mortality in the last 5 years; (2) years of schooling among household members 10 years and above; (3) school attendance among school-age children; (4) type of cooking fuel; (5) toilet type; (6) type/source of main drinking water; (7) availability of electricity; (8) type of floor material; and (9) ownership of a mobile phone, computer, or car/truck.

ART, antiretroviral therapy; BMI, body mass index; SD, standard deviation; TB, tuberculosis.

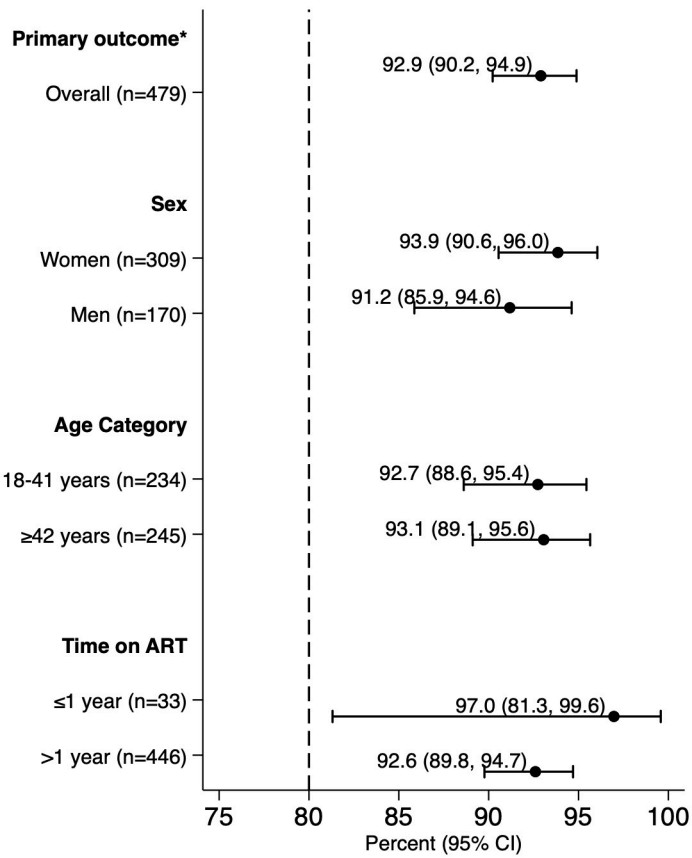

**Fig 2. 3HP acceptance and completion, by subgroup.** The forest plot shows the proportion and 95% CIs of participants accepting and competing 3HP treatment (took at least 11 of 12 doses of 3HP within 16 weeks of randomization) overall and by sex, age, and time on ART. ART, antiretroviral therapy; CI, confidence interval.
* Accepting and completing treatment (≥11 of 12 doses within 16 weeks).

## Primary outcome

In total, 445 (92.9%, 95% CI: 90.2 to 94.9) participants accepted 3HP and completed at least 11 of 12 doses within 16 weeks (Fig 2). The mean length of follow-up among those completing treatment was 78.4 days (SD: 3.5 days) and ranged from 72 days to 109 days. The longest treatment interruption among any participant reaching the primary outcome was 28 days. Treatment acceptance and completion was similarly high among men and women, younger (18 to 41 years) and older (≥42 years) participants, and participants on ART for shorter (≤1 year) and longer (>1 year) durations (Fig 2). The posterior probability that treatment acceptance and completion would exceed 0.8 under at least 1 of the 3 delivery strategies was estimated to be 99.9%.

## Secondary outcomes

A total of 34 (7.1%) participants did not complete 3HP treatment. Treatment was permanently discontinued due to a suspected 3HP-related AE in 8 (1.7%) participants (4 with pruritic rash, 1 with venous thromboembolism in the right leg, 1 with flu-like syndrome and peripheral neuropathy, 1 with pulmonary embolism, and 1 with acute liver injury—see S2 Text for additional details). In addition, 2 (0.4%) participants did not initiate treatment, 21 (4.4%) were lost to

follow-up (did not take at least 11 doses of 3HP within 16 weeks), 1 (0.2%) had treatment discontinued after becoming pregnant, 1 (0.2%) had treatment discontinued due to starting new medications with known 3HP drug interaction (clarithromycin, omeprazole, and domperidone), and 1 (0.2%) died in a motor vehicle accident.

## Discussion

In this interim analysis of an ongoing randomized trial among PLHIV in Uganda, we found extremely high acceptance and completion of 3HP with facilitated delivery strategies that all included dosing reminders, streamlined clinic visits, and reimbursement of transport costs. This treatment was also well tolerated, with less than 2% of PLHIV discontinuing treatment due to a documented AE. Taken together, these results demonstrate high levels of 3HP uptake and completion can be achieved in the context of routine HIV/AIDS care with delivery strategies designed to address known barriers to 3HP completion.

Mounting evidence of the benefits of shorter and safer rifapentine-containing regimens [7,22,23] has led to recommendations by WHO and others to implement 3HP for global scale-up of TPT. In low-burden and/or high-income countries, 3HP completion has been reported to be similar or higher in routine healthcare settings than rates reported in controlled clinical trials [24–27]. The initial Phase IV trial of 3HP, however, demonstrated poor treatment completion in its only participating African site, particularly when self-administered (38% to 50%) [10]. Recently, Churchyard and colleagues reported that treatment completion was 90.4% for self-administered 3HP versus 50.5% for 6 months of self-administered daily isoniazid in a randomized trial among PLHIV in South Africa, Ethiopia and Mozambique [28]. This trial extends those latter findings, demonstrating that similar (if not higher) levels of treatment completion can be achieved with routine healthcare providers overseeing 3HP counseling, AEs monitoring, dosing, and adherence monitoring. These data support the recent change in the Uganda HIV prevention and treatment guidelines to recommend 3HP as an alternative to isoniazid preventive therapy [29] and may encourage similar policy decisions in other high HIV/TB burden settings.

Importantly, the results reported here were achieved with the use of facilitated delivery strategies. The strategies were designed using an implementation science-based approach to target key barriers to 3HP acceptance and completion, and local stakeholders were involved in the design process to maximize acceptability and scale up potential [14–17]. The final analysis of results from the full trial is needed to determine whether one of the strategies is more or less effective than the others. However, results from this interim analysis suggest that, when scaling up 3HP, programs should consider facilitation strategies that meet the needs and preferences of PLHIV and HIV care providers in sub-Saharan Africa [30].

This interim analysis also demonstrates how data from an ongoing randomized trial can be used to inform a time-sensitive public health question while maintaining blinding of investigators to differences between arms. Further discussion is warranted among the scientific community as to when and how interim analyses of randomized trials—particularly trials focused on improving implementation of proven interventions—should be performed to inform public health decisions.

Our analysis has some limitations. First, we only assessed outcomes aggregated across trial arms. Arm-specific details have not been provided to maintain the integrity of the ongoing trial; full trial results including comparison by arm will be presented at trial completion. Nevertheless, even when conservatively assuming that completion rates are similar across arms, our Bayesian analysis suggests a greater than 99% probability of exceeding 80% initiation and

completion under at least 1 delivery strategy. Second, the trial was conducted at a single HIV/ AIDS clinic in Uganda. However, the setting is similar to the US President's Emergency Plan for AIDS Relief (PEPFAR)-funded HIV/AIDS clinics in other countries in sub-Saharan Africa, where similar barriers to completion of TPT have been widely reported [31–38]. Third, some of the facilitation strategies evaluated as part of the 3HP Options Trial are not likely to be part of usual care at most HIV/AIDS clinics in sub-Saharan Africa and would require additional funding to implement as part of 3HP scale-up. However, with increasing access to smartphones and internet in resource-limited settings, there is need for creative and inexpensive innovations to leverage this technological development to enhance health service delivery.

In conclusion, we observed extremely high acceptance and completion of 3HP for prevention of TB when delivered in a facilitated manner (i.e., with dosing reminders, streamlined visits, and reimbursement of transport costs) among PLHIV in Uganda. These interim results lend evidence to donors, policymakers, and program officials supporting the scale-up of 3HP in Uganda and elsewhere and provide some of the first data describing high 3HP completion in a high-burden setting. The findings also highlight the importance of delivering 3HP using a patient-centered approach to optimize treatment completion and thus impact on reducing the burden of TB disease and mortality among PLHIV.

## Supporting information

**S1 CONSORT Checklist. CONSORT 2010 checklist of information to include when reporting a randomized trial.** CONSORT, Consolidated Standards of Reporting Trials. (DOC)

**S1 StaRI Checklist. StaRI: The StaRI checklist for completion.** StaRI, Standards for Reporting Implementation Studies. (DOCX)

**S1 Data. Raw deidentified data used to conduct this analysis.** Each row corresponds to a trial participant. The dataset includes 14 columns: Enrollment date; Age; Sex; BMI; On ART; Years on ART; Prior TB; One-way travel time to clinic (hours); Education level (none, primary, secondary, postsecondary (university or graduate school), or vocational/tertiary); Employment status (not employed, self-employed, temporary/informal work, hired worker, or student); MPI category (severely multidimensionally poor, multidimensionally poor but not severely, not poor but vulnerable to becoming poor, or not poor nor vulnerable to becoming poor); Outcome date; Outcome reason (completed 3HP treatment, missed 6 or more doses, stopped 3HP due to a documented AE, never initiated 3HP treatment, pregnancy, new medication with 3HP drug–drug interaction, other reason, or specify (died in a motor vehicle accident); Primary outcome ($\geq$11 of 12 doses within 16 weeks of randomization). AE, adverse event; ART, antiretroviral therapy; BMI, body mass index; MPI, Multidimensional Poverty Index; TB, tuberculosis. (CSV)

**S1 Statistical Analysis Plan. 3HP Options Trial: Interim analysis statistical analysis plan.** (DOCX)

**S1 Trial Protocol. Options for delivering isoniazid–rifapentine (3HP) for TB prevention (3HP options implementation trial).** TB, tuberculosis. (DOCX)

**S1 Text. Side effects and symptom-based TB screening checklist.** TB, tuberculosis. (DOCX)

**S2 Text. Serious AEs summary.** AE, adverse event.
(DOCX)

## Acknowledgments

The authors are grateful to the administration, staff, and patients at the Makerere University Joint AIDS Program Mulago ISS clinic for their time and participation. We also thank the Infectious Diseases Research Collaboration, the Uganda TB Implementation Research Consortium (U-TIRC), and the Uganda National Tuberculosis and Leprosy Programme for supporting the study.

## Author Contributions

**Conceptualization:** Fred C. Semitala, Noah Kiwanuka, Achilles Katamba, David W. Dowdy, Adithya Cattamanchi.

**Data curation:** Jillian L. Kadota, Allan Musinguzi, Juliet Nabunje, Fred Welishe, Anne Nakitende, Lydia Akello, Opira Bishop.

**Formal analysis:** Jillian L. Kadota, Patrick P. J. Phillips, Noah Kiwanuka.

**Funding acquisition:** Adithya Cattamanchi.

**Investigation:** Fred C. Semitala, Amanda Sammann, Payam Nahid, Robert Belknap, Moses R. Kamya, Margaret A. Handley, Patrick P. J. Phillips, Anne Katahoire, Christopher A. Berger, Noah Kiwanuka, Achilles Katamba, David W. Dowdy, Adithya Cattamanchi.

**Methodology:** Fred C. Semitala, Jillian L. Kadota, Devika Patel, Amanda Sammann, Patrick P. J. Phillips, Anne Katahoire, Christopher A. Berger, Noah Kiwanuka, David W. Dowdy, Adithya Cattamanchi.

**Project administration:** Fred C. Semitala, Jillian L. Kadota, Allan Musinguzi.

**Supervision:** Fred C. Semitala, Allan Musinguzi, Achilles Katamba, David W. Dowdy, Adithya Cattamanchi.

**Visualization:** Jillian L. Kadota.

**Writing – original draft:** Fred C. Semitala, Jillian L. Kadota.

**Writing – review & editing:** Fred C. Semitala, Jillian L. Kadota, Allan Musinguzi, Juliet Nabunje, Fred Welishe, Anne Nakitende, Lydia Akello, Opira Bishop, Devika Patel, Amanda Sammann, Payam Nahid, Robert Belknap, Moses R. Kamya, Margaret A. Handley, Patrick P. J. Phillips, Anne Katahoire, Christopher A. Berger, Noah Kiwanuka, Achilles Katamba, David W. Dowdy, Adithya Cattamanchi.

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
