## [Editor Report · Decision Letter 0]

26 Aug 2021

Dear Dr Cattamanchi, 

Thank you for submitting your manuscript entitled "Completion of Isoniazid-Rifapentine (3HP) for tuberculosis (TB) prevention among people living with HIV (PLHIV): interim analysis of the 3HP Options Trial" for consideration by PLOS Medicine.

Your manuscript has now been evaluated by the PLOS Medicine editorial staff and I am writing to let you know that we would like to send your submission out for external peer review.

Please re-submit your manuscript within two working days, i.e. by Aug 30 2021 11:59PM.

Kind regards,

Callam Davidson

Associate Editor

PLOS Medicine

---

## [Decision Letter · Decision Letter 1]

7 Oct 2021

Dear Dr. Cattamanchi,

Thank you very much for submitting your manuscript "Completion of Isoniazid-Rifapentine (3HP) for tuberculosis (TB) prevention among people living with HIV (PLHIV): interim analysis of the 3HP Options Trial" (PMEDICINE-D-21-03667R1) for consideration at PLOS Medicine. 

Your paper was evaluated by an associate editor and discussed among all the editors here. It was also discussed with an academic editor with relevant expertise, and sent to independent reviewers, including a statistical reviewer. The reviews are appended at the bottom of this email and any accompanying reviewer attachments can be seen via the link below:

[LINK]

In light of these reviews, I am afraid that we will not be able to accept the manuscript for publication in the journal in its current form, but we would like to consider a revised version that addresses the reviewers' and editors' comments. You will understand that we cannot make any decision about publication until we have seen the revised manuscript and your response, and we plan to seek re-review by one or more of the reviewers. 

We hope to receive your revised manuscript by Oct 28 2021 11:59PM. Please email us (plosmedicine@plos.org) if you have any questions or concerns.

We look forward to receiving your revised manuscript and please get in touch if you have any questions. 

Sincerely,

Callam Davidson, 

PLOS Medicine

cdavidson@plos.org

As a general comment, please consider toning down your statement of generalisability to other programmatic settings (considering the single country setting and investment that would be required to make the interventions more widely available).

Please revise your title to indicate that this is an implementation science study. Your title should begin with with main concept if possible. Please place the study design (e.g. "a randomized hybrid type 3 effectiveness-implementation trial”, or similar) design in the subtitle (i.e., after the colon). 

Please report your abstract according to CONSORT for abstracts, following the PLOS Medicine abstract structure (Background, Methods and Findings, Conclusions) http://www.consort-statement.org/extensions?ContentWidgetId=562v

Abstract: 

• Please combine the Methods and Findings sections into one section, “Methods and findings”.

• Please specify who was blinded to the intervention arms.

• Please indicates the dates during which study enrollment and follow up occurred. 

• Please include the main outcome measures.

• In the last sentence of the Abstract Methods and Findings section, please describe the main limitation(s) of the study's methodology.

Please update the section heading ‘Background’ to ‘Introduction’

Please provide a definition of hybrid trials and your justification for using a hybrid type 3 effectiveness implementation design, as some readers may not be familiar with the term

Please differentiate your clinical and implementation outcomes. It will be helpful to specify implementation outcomes using standard Implementation Science terminology such as e.g., acceptability, sustainability. This tool may be of use https://cpb-us-w2.wpmucdn.com/sites.wustl.edu/dist/6/786/files/2017/08/DIRC-implementation-outcomes-tool-dg_7-27-17_ab-27xbrka.pdf

Please clearly specify and report your implementation strategies. Consider using the guidelines published by Proctor et al. to improve your reporting: Proctor EK, Powell BJ, McMillen JC: Implementation strategies: recommendations for specifying and reporting. Implement Sci 2013, 8:139

Please provide the names of the institutional review boards that provided ethical approval.

Please define the length of follow up (eg, in mean, SD, and range).

Citations should be in square brackets, please update throughout.

Please complete the CONSORT checklist and ensure that all components of CONSORT are present in the manuscript.

In addition to the CONSORT checklist please ensure that your implementation research is reported according to Standards for Reporting Implementation Studies statement (STARI). The STARI guidelines can be found here: https://www.equator-network.org/reporting-guidelines/stari-statement/

In accordance with ICMJE requirements, PLOS Medicine requires prospective, public registration of a data sharing plan (as part of mandatory clinical trials registration) for all clinical trials that began enrollment on or after January 1, 2019.

Please include the study protocol document and analysis plan, with any amendments, as Supporting Information to be published with the manuscript if accepted.

Comments from the reviewers:

Reviewer #1: "Completion of Isoniazid-Rifapentine (3HP) for tuberculosis (TB) prevention among people living with HIV (PLHIV): interim analysis of the 3HP Options Trial" describes interim results for a randomized trial involving three arms, each investigating different therapy strategies: facilitated directly observed therapy (DOT), facilitated self-administered therapy (SAT), or informed choice between DOT and SAT. From 479 trial participants (approximately equally divided between the arms) having exited the treatment period, it was found that some 93% had completed the 3HP treatment, under the definition of having at least 11 of 12 doses taken within the 16-week period. This was analyzed as having a greater than 99% probability, that treatment acceptance and completion would exceed 80% under at least one of the three strategies.

The prompt addressing of curable TB is an important issue with clear benefits in mitigating death, especially amongst individuals with HIV. The 3HP randomized trial as described appears a significant contribution towards assessing treatment programmes in an African context. Additionally, the underlying raw data has also been provided as supplementary material. However, some reservations might be addressed:

1. The manuscript as it stands is fairly concise and focused. However, while the importance of the trial should not be understated, it is unclear as to whether an official publication on interim results is strictly warranted, especially as the trial is expected to complete by October 2022 (Line 103). It might be considered to support more strongly the case for publishing interim results (e.g. the trial qualifying for early stopping, in which case the analysis supporting early stopping might be presented; early stopping however does not appear to be catered for in the trial protocol, cited as [10])

2. More details or relevant citations on the Bayesian inference framework used (currently briefly described under the Statistical Analysis section from Line 134) for the main analysis, would be illuminating.

3. While Figure 2 contains primary outcome results stratified by subgroups, there does not appear analysis relating to delivery strategies (although existing results do indeed suggest that the various strategies likely result in treatment completion rates well in excess of 0.80). The Bayesian inference analysis framework might thus be explained in greater detail, and results also presented by treatment strategy (possibly also stratified by the existing subgroups in Figure 2)

4. Related to the above, the demographic characteristics in Table 1 might also be further stratified by treatment strategy, for completeness.

Reviewer #2: This manuscript presents an interim analysis of outcome data aggregated across arms of the 3HP Options trial. While the trial's primary objective is to compare different methods of delivering 3HP, the interim analysis presents treatment completion and adverse event outcomes for patients in all three arms together. A very high proportion of patients accepted and completed treatment. Given that this interim analysis does not address the question of how best 3HP can be delivered in a high-burden setting, its importance is framed as providing one of the first examples of 3HP being delivered in a programmatic setting in a high-burden country. While I agree that publishing such examples is very important, I also think that the way the authors frame this report perhaps oversimplifies the situation; my concerns on this matter are described in comments 1 and 2 below. 

Specific comments:

1) The argument that this publication reflects treatment completion achieved in a programmatic setting in a high-burden setting is key to the authors' justification of its importance. However, this is a randomized trial being supported by the US NIH. I realize that "programmatic setting" is not a strictly defined term and exists on a spectrum where research is embedded within programs. But if the study paid for the digital adherence technology, staff monitoring treatment, or transport reimbursements, then it is hard to see how this is any more programmatic than the contexts of the 3HP efficacy trials, which had many sites in high-burden countries. If the authors want to make this distinction between their trial and previous trials, then I think further description is necessary about what the study supported and how this is more programmatic than previous trials.

2) Lines 91-92: I agree that data on acceptance and completion of 3HP in high-burden programmatic settings is sparse. However, I am not sure it is so sparse as to warrant omitting references to any examples or data from these settings. China has been using a twice-weekly version of 3HP programmatically (experience is reviewed in Cui et al. Management of latent tuberculosis infection in China: exploring solutions suitable for high-burden countries; other papers have been published reporting data from specific centers or studies). A program in Pakistan has been using 3HP and published on its experience (Yuen et al. Cost of delivering 12-dose isoniazid and rifapentine versus 6 months isoniazid for tuberculosis infection in a high-burden setting). The iAdhere trial whose US-specific results are mentioned in the preivous sentence incidentally also included a site in South Africa.

3) Could the authors provide a sentence or two more to describe the type of digital adherence technology used? I realize that this may be in the previous protocol paper, but since the message of this paper is that supportive systems of delivering 3HP can achieve high completion rates in a high-burden setting, I think it is important to understand exactly what those supportive systems are.

4) How was adverse event monitoring conducted? Verbal symptom screen? Was any routine laboratory monitoring performed? What evaluations were performed if patients reported signs/symptoms of a potential adverse event? I ask these questions because the secondary outcomes section reports the proportion of patients who had AE-related discontinuations, which is a very useful piece of information for programs. A description of AE monitoring would both help the reader assess the proportions being reported and also provide helpful information on how programs in high-burden settings can effectively monitor for AEs. Incidentally, if no routine laboratory monitoring was performed, it would be great if the authors could say this - one of the barriers frequently encountered to scaling up TB infection treatment in LMIC is the argument that the health system does not have the capacity for the type of LFT monitoring performed in high-income settings.

6) In the interest of making important information available to programs in high burden countries, it would be useful to know what the suspected 3HP-related adverse events were and their severity.

7) The discussion is rather abbreviated and does not delve much into discussing the supported self-administered mode of 3HP delivery, which I think is the most interesting aspect of this publication. Ultimately, the study reports an expected finding of high 3HP completion, which is consistent with the evidence from other trials, the substantial literature published by programs in low-burden settings, and the limited programmatic experience published to date from high-burden settings. To me, the value of this particular study is knowing how a program in a high-burden setting was able to achieve this high completion without DOT. Discussing this experience and comparing it to other experiences with digital adherence technology, which have been something of a mixed bag in terms of real-world effectiveness, would add great value to the paper.

Reviewer #3: The paper reports an interim analysis from a trial (3HP Options Trial) assessing the effect off three facilitated strategies of delivering 3HP regimen on treatment completion. The three study arms are facilitated DOT, facilitated SAT (including use of a digital adherence technology), and patient choice (with the assistance of a decision aid) between facilitated DOT and facilitated SAT. All participants receive automated dosing or clinic visit reminders weekly and are reimbursed transport costs for each clinic visit (11 visits for DOT & 2 for SAT).

The endpoint is treatment completion defined as accepting 3HP treatment and completing at least 11 of the 12 weekly doses within 16 weeks of enrolment. A Bayesian analysis was conducted to identify the posterior probability that at least one of the three arms will have treatment completion >80%. The analysis was based on participants randomised to 30 April 2021 with the target sample size expected to be completed in October 2022.

Introduction

1. The justification for this analysis was based on the urgent need for data in this in the context to aid in the global 3HP scaleup, particularly in Uganda. It is not clear how this aligns with Uganda policy on 3HP.

Methods

2. What is target sample size, expected to be completed in October 2022?

3. Is the trial design assessing for superiority or non-inferiority of study arms?

4. It is not clear what it meant by hybrid type 3 effectiveness-implementation design

5. How does this interim analysis relate to requests from external governance bodies such as a data safety monitoring board?

6. It is not clear how treatment completion is assessed in the three arms (doses taken at clinic visit, pill counts and /or DAT server data? ) - is the same method used across the three arms ?

7. Why is outcome of "initiating 3HP and completing at least 11 of the 12 weekly doses within 16 weeks of enrolment chosen"?

Results

8. How does the sample size in this interim analysis compare with the target sample size (expected to be completed in October 2022)?

Discussion

9. The authors concluded that "Taken together, these results demonstrate high levels of 3HP uptake and completion can be achieved in a programmatic setting when delivery strategies are designed to address known barriers to TPT completion." Given the intervention arms have varying degrees of support and therefore varying costs attached, how does this issue contribute to faster roll-out of 3HP in the region. Would be interesting for the authors to comment on whether the costing of the interventions would also influence the type of intervention to be rolled-out? Which of the three interventions will the authors be recommending to the Ministry of Health in Uganda?

10. The authors report on the recently completed trial in South Africa, Ethiopia and Mozambique which found high levels of treatment completion achieved in a programmatic context. They continue by saying that their study highlights the value of facilitated delivery deliveries - though in reality we do not know the effects of the different interventions as the interim analysis combined data across the three arms.

11. It would be useful for the authors to expand on their assertion that this ongoing analysis helped inform time critical public health questions. Are the authors recommending this type of analysis for similar scenarios?

Reviewer #4: This manuscript details an interim analysis of the ongoing 3HP Options Trial in Uganda, a randomized trial of DOT vs. SAT vs. SAT+DAT that is evaluating acceptance and completion rates across these three treatment strategies for TPT in PLHIV. Considering the need for more evidence for acceptance and adherence of these optimized strategies in the context of routine HIV care in sub-Saharan Africa, this manuscript will directly and substantially support the WHO's recommendation for using 3HP as an option for TPT in this setting.

The authors describe an inference analysis, estimating that at least one optimized treatment strategy would lead to >80% acceptance and completion of 3HP. Following their analysis, the authors demonstrated that all three strategies had a very high probability of exceeding 80% acceptance and treatment completion, effectively supporting that all three optimized strategies are highly effective in a sub-Saharan Africa context in Uganda.

The findings of this analysis are in context of previous literature on the topic, and the data analysis clearly supports the findings presented.

This manuscript details an interim analysis of an ongoing randomized trial and details of trial design have been previously published and are accessible via PubMed. 

The overall study conforms to CONSORT guidelines for a randomized trial, and the methodology for the analysis to be reproduced.

Overall, the manuscript is well written, is accessible to non-specialists, and the findings will add to the body of evidence supporting high acceptability and completion rates for 3HP for TPT, using both DOT and SAT approaches, in the context of routine HIV care in sub-Saharan Africa.

Line-by-line minor comments, questions:

Line 90 - consider emphasizing that the Belknap study demonstrated that SAT was noninferior to DOT, but only specifically in the United States component of this study (further supports the authors' argument that evidence on 3HP acceptability and completion rates from studies conducted outside the United States, especially in SSA, is needed.

114 - is the digital adherence technology (DAT) text based? Consider adding explanation on what the participant was required to do with DAT. IVR is clearly explained in the manuscript.

129 - per the primary outcome: "the proportion of randomized participants who accept 129 3HP treatment and complete at least 11 of the 12 weekly doses within 16 weeks of enrollment." Did any participants who reached this outcome have a treatment interruption of 4 weeks before resuming weekly dosing and completing 3HP? I ask this as professional opinion can vary on the acceptable maximum treatment interruption allowed before a patient must restart 3HP (in my experience, between 1 and 4 weeks). Consider including the range of treatment interruption (in days) that was observed by this analysis for individuals who did successfully complete 3HP.

150 (Figure 1) - one 'did not meet inclusion criteria' in figure 1 states: "resistance to isoniazid (INH) or rifapentine (RPT) (n=2)". How was this determined/known? Were these individuals previously treated for active TB disease and now qualified for TPT? Seems that would not be the case as another 'did not meet inclusion criteria' is "prior treatment for TB/preventive TB". Please clarify.

179 - considering describing most common adverse event(s) that led to 1.7% of participants discontinuing treatment.

182 - unless it represents an increased risk for identifying the participant, consider including the name of the medication with a known 3HP interaction.

The Discussion section does not include limitations. I encourage you to discuss some possible limitations of your analysis. One plausible limitation is that this is an interim analysis of 3HP uptake for an ongoing study in Kampala, Uganda, and the acceptability and treatment completion rates observed in this setting may not reflect the probability of acceptance and completion of 3HP in other countries and settings in sub-Saharan Africa.

Overall, a well written manuscript. My recommendation is 'accept with minor revision'.

Clay Roscoe, MD MSc

HIV Services, Family Medicine Residency of Idaho, Boise, Idaho, USA

CMO, Sedia Biosciences, Beaverton, OR, USA

Reviewer #5: The authors present an interim analysis of limited data from the pragmatic randomized controlled trial, 3HP Options Trial, of the acceptance and completion (combined as a single outcome) of 3 months of weekly isoniazid and rifapentine (3HP) as TB preventive therapy (TBPT) in people living with HIV (PLHIV). The 3HP Options Trial aims to compare the acceptance, completion and costs (among other outcomes) of 3 different 3HP delivery strategies. The trial is still enrolling and the outcome data from all three arms were combined and included as an aggregate for this analysis. There is no comparator group.

TBPT can prevent TB disease in PLHIV and practical alternatives to 6 months of daily isoniazid (which is standard of care in many regions) make an important contribution to public health. 3HP has been shown to be as effective as isoniazid for this purpose and is recommended by the WHO. As the authors note, uptake in previous clinical trials has varied between 38-50% and 90%. In this analysis, the authors report high acceptance and completion of 3HP as TBTP (92.9%), and calculate that this (i.e. acceptance and completion) would exceed 80% if one of the three treatment strategies in the trial were used in this setting. These results did not vary by sex, age or time on antiretroviral therapy.

The primary outcome/s of the 3HP Options Trial will hopefully indicate a best practice for dispensing 3HP to PLHIV in Uganda. Other elements of the study design may provide a template for stakeholder and community engagement and support patient-centred care. The aggregation of the data in these analyses limits the novelty and usefulness of the results. In addition, as presented, the description of the settings and methods do not support the conclusion that the results are generalizable to a programmatic setting.

The manuscript is very well-written however, the impact is overstated. It may be better suited to an alternative publication. Given the remit that 'papers published in PLOS Medicine should be of the highest technical quality and be seen as significant advances within their own discipline and beyond' I do not think this interim analysis is suitable for publication in this journal.

Major comments

While the 3HP Options Trial may be implemented in a routine clinical setting, the trial study procedures are not routine and vary from intensive observed monitoring (with re-imbursement for visits) to digital adherence monitoring and automated dosing and clinic reminders. The implication that the results have been achieved in a programmatic context (line 198) is that they are generalizable to similar routine clinical settings in Uganda and elsewhere in sub-Saharan Africa. Such practices described above are not routine in most programs which are often under-resourced. 

The authors reference the 3HP Options Trial protocol but do not describe the setting, including physical, technological and human resources. Did clinic staff receive incentives for participating in the trial management? What was the source of funds for re-imbursement? At what level was the political and management support provided? Related to the lack of detail in the methodology, streamlined clinic visits and other patient-centred (line 202) and facilitated delivery strategies (line 186) are not described but are alluded to in the discussion. In addition, one of the strengths of the study design is noted to be the engagement of relevant stakeholders (210), but this does not appear elsewhere in the manuscript (methods or results).

Minor comments

Is the trial registered?

Study design: The power calculations have been based on the parent study which aims to enrol 1656 adults. Approximately 30% of the target have been enrolled. Is this sufficient for the interim analysis to be meaningful? 

As noted above, the background planning and the integration of the study procedures into the routine setting are not well described. This is necessary if the conclusion drawn is that the systems work in a program setting.

[LINK]

---

## [Decision Letter · Decision Letter 2]

17 Nov 2021

Dear Dr. Cattamanchi,

Thank you very much for re-submitting your manuscript "Completion of Isoniazid-Rifapentine (3HP) for tuberculosis (TB) prevention among people living with HIV (PLHIV): interim analysis of a hybrid type 3 effectiveness-implementation randomized trial" (PMEDICINE-D-21-03667R2) for review by PLOS Medicine.

I have discussed the paper with my colleagues and the academic editor and it was also seen again by five reviewers. I am pleased to say that provided the remaining editorial and production issues are dealt with we are planning to accept the paper for publication in the journal.

[LINK]

We look forward to receiving the revised manuscript by Nov 24 2021 11:59PM.   

Sincerely,

Callam Davidson, 

Associate Editor 

PLOS Medicine

plosmedicine.org

Requests from Editors:

Please remove the abbreviations "TB" and "PLHIV" from the title.

Please include the baseline summary data (from lines 251-253) in your abstract. 

Line 119: Please state the number of patients discontinued (8) as well as the percentage.

Please define AST and ALT in Figure 1.

Thank you for including a STARI checklist. Please adapt the checklist to use sections and paragraph numbers rather than page numbers (which will likely change during the revision process). Column headers can be updated to reflect this change (from 'page number' to 'Section name and paragraph number', or similar). 

To help us extend the reach of your research, please provide any Twitter handle(s) that would be appropriate to tag, including your own, your co-authors, your institution, funder, or lab.

Comments from Reviewers:

Reviewer #1: We thank the authors for acknowledging our prior concerns. The main reservation remains that this is an interim analysis with arm-specific details unable to be further provided to maintain the integrity of the ongoing trial. That said, the primary objective (i.e. a greater than 99% probability that treatment acceptance and completion would exceed 80%, under at least one of the three strategies) appear highly likely to have been attained, and the release of these interim results (as opposed to late 2023 for the full results) would appear to help guide 3HP treatment promptly. Considering this, we have in principle no objection to the release of these results if judged significant enough by the editors, particularly if the inability to reveal arm-stratified analysis is further emphasized as a limitation.

Reviewer #2: Thank you to the authors for the revision. The paper is substantially more balanced in the way it contextualizes its significance, and it is more informative for programs considering how to deliver 3HP in high-burden settings. I agree with the other reviewers who have questioned the importance of an interim results publication that neither answers the question of how best 3HP can be delivered nor discusses the implementation experience with these methods (presumably saving both for what will be the "main" trial publication). However, I think that on the balance, the information that is now presented about the use of 99DOTS, adverse event monitoring under programmatic conditions, and the use of transport reimbursements and other patient support strategies, in the context of overall high treatment completion, gives sufficient value to warrant more prompt dissemination.

The one comment I have is the added citation of the RE-AIM framework. I am sure that the larger study was indeed designed using the RE-AIM framework, but it is difficult to see how this interim report maps to it. It is not clear from the description of the trial that 3HP is the intervention and DOT/99DOTS are the implementation strategies, so it is hard to intuitively see treatment completion as a fidelity measure since it is often treated as a proxy effectiveness measure. In addition, it is confusing to cite use of the RE-AIM framework when only two of the components are addressed as a composite outcome, and the other three are not addressed at all (adoption and maintenance are not even mentioned among the outcomes referenced in the protocol). I suggest clarifying in the study design section that 3HP is the evidence-based intervention and that DOT vs supported SAT are the implementation strategies being assessed. And I suggest either briefly summarizing the main outcomes corresponding to the RE-AIM components or at the very least being explicit that this interim analysis addresses only a combined reach+implementation fidelity outcome, while the other RE-AIM components will be more fully assessed at the end of the trial.

Reviewer #3: Thank you for your detailed response to comments. My concerns have been addressed.

Reviewer #4: I very much agree with Reviewer #5's comment that this interim research, overall, may not meet full PLOS Medicine criteria for publication, which includes the language that publications "be seen as significant advances within their own discipline and beyond". As an interim analysis of 3HP TPT acceptance and completion, based on aggregated data across the study arms being evaluated, this manuscript may not fully fit PLOS Medicine criteria for publication. Though this interim analysis does not address the question of how best 3HP can be delivered in a high-burden setting (the authors do state the final research will address this), it does, importantly, provide one of the first examples of 3HP being delivered in a programmatic setting in a high-burden country, demonstrating positive acceptance and fidelity of 3HP, in general. For the latter findings, I recommend this manuscript be accepted for publication, but respectfully defer to the Editor's final decision on if this manuscript represents a significant advance in the field of TPT and meets journal criteria for publication.

Reviewer #5: Much more detail and clarity has been provided and the manuscript improved. The tone has been moderated and the data presented more appropriately (to my mind); my comments have been addressed. The manuscript is sound and well-written.

I am still not certain that the impact is sufficient for publication in PLOS MEDICINE, but the editors are better placed in this regard.

[LINK]

---

## [Editor Report · Decision Letter 3]

25 Nov 2021

Dear Dr Cattamanchi, 

On behalf of my colleagues and the Academic Editor, Dr Claudia Denkinger, I am pleased to inform you that we have agreed to publish your manuscript "Completion of Isoniazid-Rifapentine (3HP) for tuberculosis prevention among people living with HIV: interim analysis of a hybrid type 3 effectiveness-implementation randomized trial" (PMEDICINE-D-21-03667R3) in PLOS Medicine.

When making the required changes, please also remove the 'Funding' and 'Conflicts of interest' sections from your title page (both will be published as metadata based on your response to the relevant submission form question).

PRESS

Sincerely, 

Callam Davidson 

Associate Editor 

PLOS Medicine